# The Association between Prenatal Per- and Polyfluoroalkyl Substances Exposure and Neurobehavioral Problems in Offspring: A Meta-Analysis

**DOI:** 10.3390/ijerph20031668

**Published:** 2023-01-17

**Authors:** Huojie Yao, Yingyin Fu, Xueqiong Weng, Zurui Zeng, Yuxuan Tan, Xiaomei Wu, Huixian Zeng, Zhiyu Yang, Yexin Li, Huanzhu Liang, Yingying Wu, Lin Wen, Chunxia Jing

**Affiliations:** 1Department of Public Health and Preventive Medicine, School of Medicine, Jinan University, No. 601 Huangpu Ave West, Guangzhou 510632, China; 2Guangzhou Center for Disease Control and Prevention, Guangzhou 510440, China; 3Guangdong Key Laboratory of Environmental Exposure and Health, Jinan University, Guangzhou 510632, China

**Keywords:** PFAS, ADHD, ASD, pregnancy, offspring

## Abstract

Exposure to per- and polyfluoroalkyl substances (PFAS) during pregnancy has been suggested to be associated with neurobehavioral problems in offspring. However, current epidemiological studies on the association between prenatal PFAS exposure and neurobehavioral problems among offspring, especially attention deficit/hyperactivity disorder (ADHD) and autism spectrum disorder (ASD), are inconsistent. Therefore, we aimed to study the relationship between PFAS exposure during pregnancy and ADHD and ASD in offspring based on meta-analyses. Online databases, including PubMed, EMBASE, and Web of Science, were searched comprehensively for eligible studies conducted before July 2021. Eleven studies (up to 8493 participants) were included in this analysis. The pooled results demonstrated that exposure to perfluorooctanoate (PFOA) was positively associated with ADHD in the highest quartile group. Negative associations were observed between perfluorooctane sulfonate (PFOS) and ADHD/ASD, including between perfluorononanoate (PFNA) and ASD. There were no associations found between total PFAS concentration groups and neurobehavioral problems. The trial sequential analyses showed unstable results. Our findings indicated that PFOA and PFOS exposure during pregnancy might be associated with ADHD in offspring and that prenatal PFOS and PFNA exposure might be associated with ASD in offspring. According to the limited evidence obtained for most associations, additional studies are required to validate these findings.

## 1. Introduction

Perfluoroalkylated substances (PFASs) are forms of persistent organic pollutants, which have the characteristics of bioaccumulation, long-distance migration, and high toxicity, and they have adverse effects on the environment and human health [1,2,3]. Since the 1950s, PFASs have been widely used in various commercial products, including textile, paper, tableware coatings, food packaging, carpets, anti-fouling agents, and other forms of production and life [4,5,6]. PFAS and their terminal degradation products can be stable in soil and water for over 7 years [7,8]. Currently, PFASs have been detected in various environmental media of human lives, such as the atmosphere, water, and soil [9,10,11], including animal and plant-derived foods [12,13,14].

The National Biomonitoring Program from CDC states that at least one PFAS can be detected in blood samples among the U.S. population [15], and numerous evidence indicates that PFAS can also be detected in pregnant women’s bio samples [16,17,18]. Moreover, the developing offspring can be affected through the placenta and breastfeeding [19]. Present studies show that PFASs can cross the placenta, and they are further detected in the umbilical cord blood and amniotic fluid [20,21,22,23,24,25,26,27,28,29]. The placenta is a plausible target for PFASs [30], which may harm the developing embryo [31,32]. Animals and in vitro studies have also revealed a series of neurotoxins [33,34,35,36,37]. However, the relationship between PFAS exposure and neurobehavioral problems, such as autism spectrum disorder (ASD) and attention deficit/hyperactivity disorder (ADHD), remains uncertain [38,39,40,41].

Current epidemiological studies on the association between prenatal PFAS exposure and neurobehavioral problems in offspring remain unclear. Several studies have reported a positive association between PFAS and the risk of ADHD and ASD [42,43,44,45], while others yielded a negative result [46,47,48,49]. Some studies reported protective associations between PFAS and both neurobehavioral problems [29,50,51]. These results may be due to study design, reverse causality, prospective studies with different populations, and sample sizes. A larger study with more reliable cases is needed to explore these relationships. Hence, we performed a systematic meta-analysis to explore the relationship between prenatal PFAS exposure and ADHD and ASD.

## 2. Methods

### 2.1. Data Origins and Search Tactics

This research study rigorously refers to the principle of the Preferred Reporting Items for Systematic Reviews and Meta-Analyses (PRISMA) statement [52], and we systematically searched all studies related to prenatal PFAS exposure and neurobehavioral problems in offspring that were published before July 2021 in the three electronic databases: PubMed, EMBASE, and Web of Science. We used the following keywords: “perfluoroalkyl”, “polyfluoroalkyl”, “perfluorinated”, “perfluorooctanoic”, “perfluorooctance”, “perfluorohexane”, “PFOS”, “PFOA”, “PFHxS”, “PFAS *”, “Attention Deficit Hyperactivity Disorder”, “ADHD”, “hyperactivity”, “adhd”, “attention deficit”, “attention-deficit/hyperactivity”, “Autistic Spectrum Disorder”, “ASD”, “autistic disorder”, “asd”, “autistic”, “autism”, “intelligence”, “intelligence”, “IQ”, “Cognition”, “cognition”, “Comprehension”, “cognitive”, “neurocognition”, “neurocognitive”, “neuro-cognition”, “neuro-cognitive”, “Executive Function”, “executive function”, “executive functions”, “executive functioning”, “brain”, “brain function”, “brain structure”, “fMRI”, “functional magnetic resonance imaging”, “event-related potential”, “event related potential”, “Problem Solving”, “problem solving”, “Memory”, “memory”, “Attention”, “attention”, “attentiveness”, “concentration”, “concentrate”, “learn”, “learning”, “brain development”, “cognitive performance”, “cognitive function”, “cognitive functioning”, “information retrieval”, “information processing”, “perceptual skills”, and “intelligence quotient“. To avoid missing related studies, we also performed a manual search on the reference list of studies that have been searched.

### 2.2. Study Selection

The authors (HJY and YYF) separately screened all studies by titles and abstracts and ascertained the eligible studies by reading the full text. The studies that were eventually included met the following criteria: (1) observational study design, including case–control or cohort studies; (2) exposure to at least one PFAS (e.g., perfluorooctanoic acid (PFOA), perfluorooctane sulphonate (PFOS), perfluorononanoic acid (PFNA), perfluorohexane sulfonic acid (PFHxS), and perfluoroalkane sulfonamido substances); (3) measure PFAS exposure during pregnancy using maternal blood samples (e.g., serum, plasma, or whole blood); (4) the results are either prenatal neurological or child neurological as related problems; and (5) risk estimates are provided, including odds ratios (OR), relative risk (RR), or hazard ratio (HR) and 95% confidence interval (CIs). Laboratory research, non-human animal research, letters, and reviews were excluded. When the same data appear in multiple studies, only studies with the largest data sets are included in the analysis. The screening process of document screening is shown in the flowchart.

### 2.3. Definition of Outcomes

Due to the limited amount of related research, only two neurobehavioral diseases, ADHD and ASD, were taken into consideration in our study. The diagnosis of ADHD and ASD in offspring may be determined by medical institution registration, expert opinion, or questionnaire survey.

### 2.4. Data Extraction and Quality Assessment

The following elements were extracted from each eligible study: name of the first author; year of publication; country; neurobehavioral outcome; type of PFAS; the number of pregnant women; age of the pregnant women; biosample type; pregnancy week of sample collection; study time; maximum adjustment of ORs, RRs or HRs and 95% CIs; adjustment of covariates. The Newcastle–Ottawa Quality Assessment Scale (NOS) [53] was used as the guideline for evaluating the included articles. The quality assessment of the study includes selection, comparability, exposure (case–control studies), or results (cohort studies). The highest score of NOS is 9 points, and studies with a score of ≥ 7 can be considered as high-quality studies. The data extraction and quality assessment for each study were conducted independently by YHJ and FYY, and the disagreement was resolved via discussions between XQW, HJY, and YYF.

### 2.5. Statistical Analysis

In this research study, Stata software (version 16) was used for statistical analyses. We used quantitative meta-analysis when there were ≥ 3 qualified studies on the specific association between PFAS exposure in pregnancy and children’s neuro-related behaviors that contain identical types of data. A simplified summary was conducted when qualified studies were less than three. Pooled risks and 95% CIs with the fixed-effects model or the random-effects model were assessed as the main statistical method according to the heterogeneity. The heterogeneity of the articles involved was tested with the I^2^ and *p*-value, and *p*-value < 0.05 was regarded as heterogeneous, with an I^2^ statistic < 25% indicating low, 25–50% indicating moderate, and >50% indicating high heterogeneity. A random-effects model was used in cases when I^2^ ≥ 50% [54].

When there were more than 3 eligible studies, we conducted a dose–response meta-analysis using linear and non-linear models and selected the significant model with the best goodness-of-fit χ^2^ score [55]. When extracting dose data, median doses were selected. The extraction method is shown below. First, in the reference group, the dose was set at zero. Second, when the beginning of the smallest dose group is an open interval, the beginning value is replaced by zero (e.g., 0–10 if < 10). Third, if the end of the largest dose group is an open interval, the width of the previous dose group (e.g., 20−10 = 10) is added to the beginning value of the last group, and the resulting value is set as the end (e.g., 20–30 if > 20). For each dose group, the median was set as the exact dose (e.g., 5 if 0–10) [56]. Since the corresponding PFAS concentrations in each article were different, we recorded the total and quartile concentration of PFAS. For the study that only provides tertile concentrations, we used the tertile percentile as dose concentrations. The pooled odds ratios were calculated for ADHD/ASD associated with per 1 ng/mL increase in PFAS concentrations in continuous analyses and for ADHD/ASD associated with PFAS concentrations by comparing the highest category with the lowest category in quantile analyses [57].

To assess the publication bias in this study, Begg’s test and Egger’s regression asymmetry test were performed. A funnel chart was used to evaluate the results. Simultaneously, the sensitivity analysis was carried out to observe the stability of the comprehensive results.

### 2.6. Trial Sequential Analysis

The risks of random errors were tested by the trial sequential analysis (TSA) with the Stata package (metacumbounds command), aiming at maintaining an overall 5% risk of a Type I error and a power of 80%. The information size calculation for meta-analyses (cumulated sample size of included trials) was combined with an adapted threshold of statistical significance in the cumulative meta-analysis. With the control event’s proportion calculated from the actual meta-analysis in the calculation of the required information size, an intervention effect of a 20% relative risk reduction (RRR) was anticipated [58].

## 3. Results

### 3.1. Studies Included in the Meta-Analysis

A total of 1755 articles were retrieved by a literature search of PubMed, Embase, and Web of science. After the removal of 219 duplicates, 1536 studies were further screened for title and abstract. A total of 1424 articles were removed due to systematic reviews, cross-sectional studies, and other unrelated topics. We excluded 101 of 112 articles due to the unavailable concentration of PFASs during maternal pregnancy, lack of quantitative estimates, or duplicate publications. Finally, the remaining 11 articles were included in the present meta-analysis (Figure 1). Two of the eleven articles contained studies related to both ADHD and ASD, seven articles were about ADHD [42,43,44,46,47,48,50], and six articles were about ASD [29,43,45,49,50,51]. The NOS score in this meta-analysis ranged from 7 to 9, which is considered a high-quality article (Appendix A). The characteristics of this article and possible confounding factors are described in Table 1.

### 3.2. Association between PFAS and ADHD

With the systematic literature search, seven studies focusesd on pregnancy PFOA (*n* = 7), PFOS (*n* = 7), PFNA (*n* = 3), and PFHxS (*n* = 3) on ADHD were included. In the total concentration group, the summarized evidence suggests that there is no significant association between prenatal PFAS exposure and ADHD in offspring. Pooled estimates for comparisons of the highest vs. lowest quartile showed that PFOA (OR = 1.21; 95% CI:1.04–1.42; *p* = 0.016) and PFOS (OR = 0.85; 95% Cl: 0.721–0.996; *p* = 0.045) were significantly associated with ADHD (Table 2 and Table 3; Figure 2 and Figure 3). Dose–response analyses indicated that there is no linear association between pregnancy PFOA and PFOS exposure and ADHD risks in offspring (Figure 4).

### 3.3. Association between PFAS and ASD

We found six studies on PFAS exposure in pregnancy and ASD, including six studies on PFOA, six studies on PFOS, five studies on PFNA, and five studies on PFHxS. Although the pooled estimates in the total concentration group indicated that PFASs were unlikely associated with ASD, the highest concentration categories of PFOS (OR = 0.79; 95% Cl: 0.64–0.98; *p* = 0.035) and PFNA (OR = 0.80; 95%Cl: 0.65–0.98; *p* = 0.032) were negative for ASD compared to the lowest category (Table 2 and Table 3; Figure 5 and Figure 6). The dose–response analysis showed that there was no dose–response relationship between PFAS exposure and ASD risk (Figure 7).

### 3.4. Trial Sequential Analysis

The highest quartile arrays of PFOA and PFOS associated with ADHD, including the highest quartile array of PFOS and PFNA associated with ASD, were subjected to trial sequential analyses. Appendix A show the results of the trial sequential analysis. All the Z curves did not intersect the TSA boundary (*p* > 0.05), suggesting that the summary result might be unstable.

### 3.5. Publication Bias

No publication bias was found in this meta-analysis (Appendix A).

## 4. Discussion

In this systematic review and meta-analysis, we systematically gathered the available evidence on the effect of prenatal PFAS exposure with respect to the outcome of neurobehavioral problems in future generations. We included 11 articles in the present study. In the total concentration group, the relationships between PFAS (PFOA, PFOS, PFNA, and PFHxS) and ADHD or ASD could not be identified. However, in the highest quartile group, PFOA is positively associated with ADHD, while PFOS is negatively associated with ADHD and ASD. PFNA is negatively correlated with ASD when compared with the lowest quartile group. There is no publication bias in this study. Unlike the study designed by Joan et al. [60], we directly investigated the effects of maternal PFAS exposure on neurobehavioral problems in offspring, which effectively explored the relationship between PFAS exposure and neurobehavioral problems in offspring.

In the highest quartile group, PFOA exposure increased the risk of ADHD in future generations, which is consistent with the results of previous animal studies [61,62]. Pregnant CD-1 mice exposure to PFOA results in significantly higher activity levels in their male offspring [61], and fertilized zebrafish under PFOA exposure showed morphological changes and hyperactivity [62]. Our study revealed that PFNA exposure potentially decreases the risk of ASD in the offspring, and PFOS exposure potentially reduces the risk of ADHD and ASD in future generations in the highest quartile group, which is contrary to the results of some animal studies [62,63,64,65,66,67]. Several studies show that exposure to PFOS during pregnancy results in hypermobility in rat offspring [63], and hyperactivity was found in fertilized zebrafish when exposed to PFOA [62,64]. Ghassabian et al. found that PFOS exposure is positively associated with children’s behavioral difficulties [65]. The significant positive correlation between PFNA exposure and the total SDQ score was presented in the Hoyer and Luo studies, respectively [66,67]. One plausible reason for the inconsistent results may be that the doses exposed to animals and humans are different. The exposure concentrations used in animal experiments are usually higher than the human routine exposure. Animals exposed to a higher dose may experience other serious side effects such as increased mortality and birth defects, which could make the observed neurotoxicity biased [68,69].

In the total concentration group, there is no significant association between PFAS exposure and ADHD, which is consistent with many of the studies [70,71,72,73,74]. Previous studies of continuous-dose PFAS exposure have also yielded no significant results [70,71,72,73,74]. One of the possible hypotheses is that prenatal PFAS exposure and neurobehavioral problems in offspring might have a non-linear relationship. The study by Christou et al. revealed that zebrafish larval hyperactivity or underactivity was related to PFAS concentrations and the age of the zebrafish [75]. Some studies show that there are gender differences in the relationship between PFAS exposure and neurobehavioral problems [39,72,76]. Moreover, there is an uncertain threshold of PFAS exposure on human health [70,77,78]. Stein et al. found that there was an inverted “J”-shaped association between PFOA exposure and ADHD [38]. Low-dose prenatal PFOS exposure leads to an overactive basal swimming speed, but the offspring continue to be exposed after delivery, and the overactive state gradually disappears [79]. Liew et al. concluded that, assuming uncontrolled confounding and selection bias due to fetal death, the observed association between PFAS exposure and ADHD is either positive or negative [80]. This may explain the inconsistency between our results and existing studies.

There are several advantages in the present study. This is one of the first studies to comprehensively and systematically evaluate the relationship between prenatal exposure to PFAS and neurobehavioral problems in offspring. More than 8000 participants were included in this study, which makes the summary results more reliable. The use of various statistical methods also improved the effectiveness of meta-analysis. However, due to the lack of existing studies, sequential analyses suggested that the summary result might be unstable. In addition, hierarchical analyses cannot be conducted. According to the different study designs and methods, the dose–response analyses cannot be accurately assessed based on the dose–response relationship. Therefore, more in-depth studies are needed.

## 5. Conclusions

In summary, PFOA, PFOS, and PFNA exposure during pregnancy were found to be associated with ADHD and ASD in offspring in the highest quartile group. The findings indicated that prenatal PFAS exposure and neurobehavioral problems in offspring are not linear associations, but the research result is unstable. More research should be encouraged to further study its associations and mechanisms of action.

## Figures and Tables

**Figure 1 ijerph-20-01668-f001:**
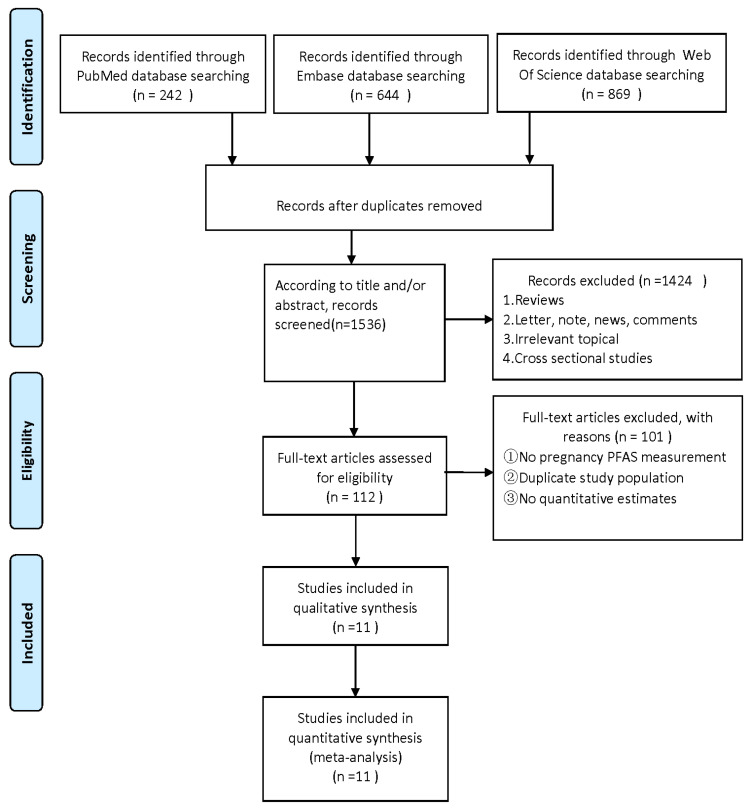
Flowchart for the screen of included studies in the current meta-analysis [59].

**Figure 2 ijerph-20-01668-f002:**
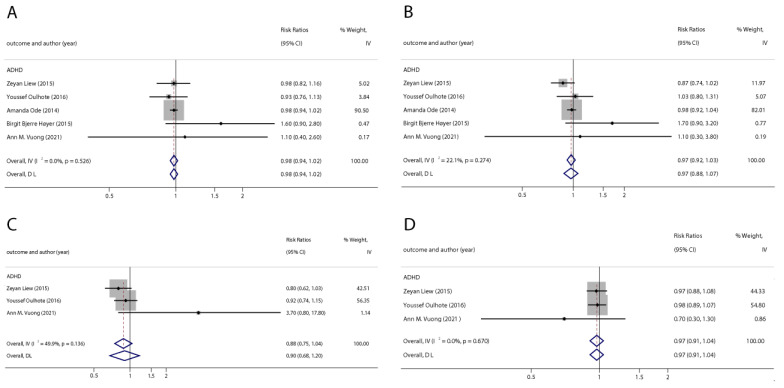
Pooled odds ratios for ADHD associated with per 1 ng/mL increase in polyfluoroalkyl substance concentrations. (**A**): PFOA; (**B**): PFOS; (**C**): PFNA; (**D**): PFHxS. Abbreviations: CI, confidence interval; PFOA, perfluorooctanoate; PFOS, perfluorooctane sulfonate; PFNA, perfluorononanoate; PFHxS, perfluorohexane sulfonate; ADHD: attention deficit hyperactivity disorder [42,44,46,47,50].

**Figure 3 ijerph-20-01668-f003:**
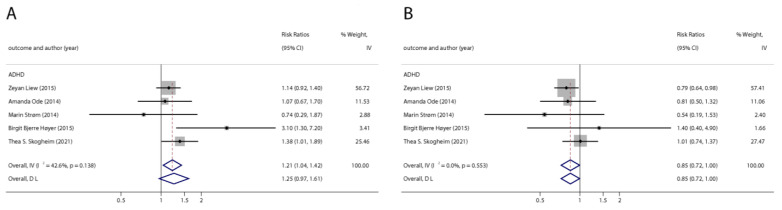
Pooled odds ratios for ADHD associated with polyfluoroalkyl substances concentrations: comparison of the highest category with the lowest category. (**A**): PFOA; (**B**): PFOS. Abbreviations: CI, confidence interval; OR, odds ratio; PFOA, perfluorooctanoate; PFOS, perfluorooctane sulfonate; ADHD: attention deficit hyperactivity disorder [42,43,47,48,50].

**Figure 4 ijerph-20-01668-f004:**
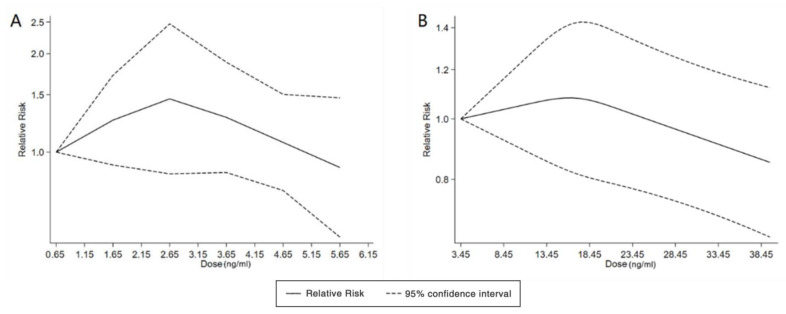
Dose–response relationship between polyfluoroalkyl substances and the risk of ADHD. (**A**): PFOA; (**B**): PFOS. Abbreviations: PFOA, perfluorooctanoate; PFOS, perfluorooctane sulfonate; ADHD: attention deficit hyperactivity disorder.

**Figure 5 ijerph-20-01668-f005:**
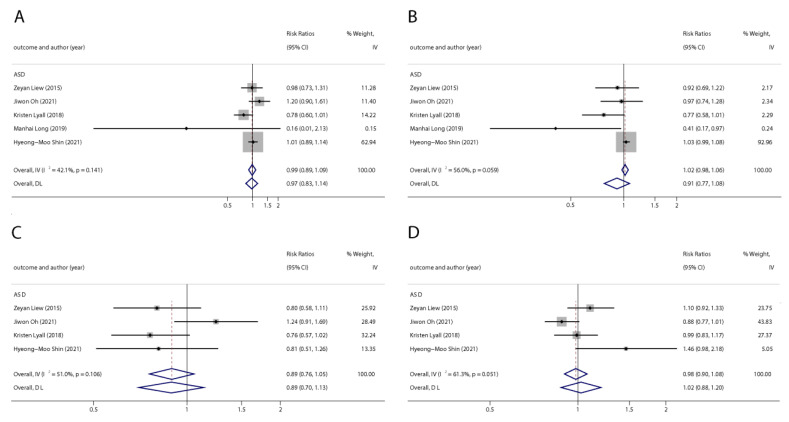
Pooled odds ratios for ASD associated with per 1 ng/mL increase in polyfluoroalkyl substances concentrations. (**A**): PFOA; (**B**): PFOS; (**C**): PFNA; (**D**): PFHxS. Abbreviations: CI, confidenceinterval; PFOA, perfluorooctanoate; PFOS, perfluorooctanesulfonate; PFNA, perfluorononanoate; PFHxS, perfluorohexane sulfonate; ASD, autism spectrum disorders [29,45,49,50,51].

**Figure 6 ijerph-20-01668-f006:**
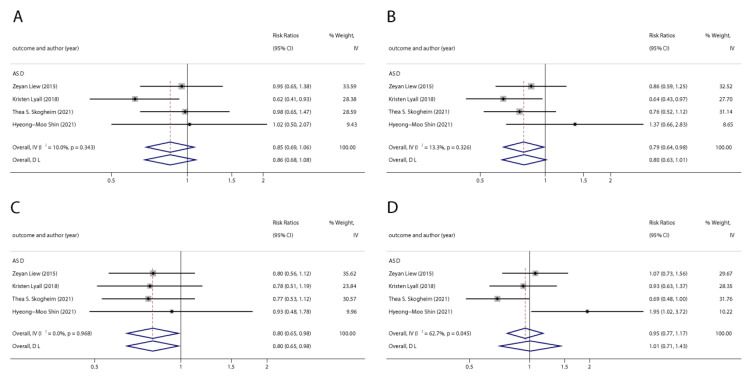
Pooled odds ratios for ASD associated with polyfluoroalkyl substances concentrations comparing the highest category with the lowest category. (**A**): PFOA; (**B**): PFOS; (**C**): PFNA; (**D**): PFHxS. Abbreviations: CI, confidence interval; PFOA, perfluorooctanoate; PFOS, perfluorooctane sulfonate; PFNA, perfluorononanoate; PFHxS, perfluorohexane sulfonate; ASD, autism spectrum disorders [43,49,50,51].

**Figure 7 ijerph-20-01668-f007:**
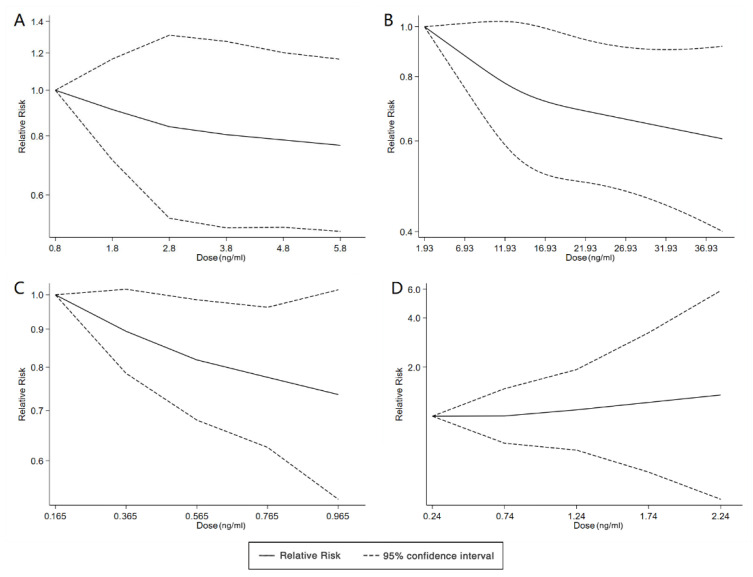
Dose–response relationship between polyfluoroalkyl substances and the risk of ASD. (**A**): PFOA; (**B**): PFOS; (**C**): PFNA; (**D**): PFHxS. Abbreviations: PFOA, perfluorooctanoate; PFOS, perfluorooctane sulfonate; PFNA, perfluorononanoate; PFHxS, perfluorohexane sulfonate; ASD, autism spectrum disorders.

**Table 1 ijerph-20-01668-t001:** The included studies on the association between prenatal PFAS exposure and neurobehavioral problems in offspring.

First Author, Year	Country	Maternal Age, Yrs	No. of Participants	PFAs, ng/ml	PFAs Measurement	Maternal Sample Type	Gestational Week of Sample Collection	Outcome (No.)	NOS Score
(Liew, Ritz et al., 2015) [50]	Danmark	Median: 28.52	990	PFOS/PFOS/PFHxS/ PFHpS/PFNA/PFDA	LC-MS/MS	Plasma	Range: the first trimester	ADHD: 220 ASD: 220	7
(Oulhote, Steuerwald et al., 2016) [46]	Danmark	Mean 29.40	656	PFOA/PFOS/PFHxS/ PFNA/PFDA	HPLC-MS/MS	Serum	Around 32	ADHD: 539	7
(Ode, Kallen et al., 2014) [47]	Sweden	Median: 27.65	412	PFOS/PFOA/PFNA	HPLC-MS	Serum	Born	ADHD: 206	9
(Strøm, Hansen et al., 2014) [48]	Danmark	Median: 27.97	876	PFOA/PFOS	LC-MS/MS	Serum	Around 30	ADHD: 27	8
(Hoyer, Ramlau-Hansen et al., 2015) [42]	Ukraine and Poland	Mean: 26.00 range: 20–35	1106	PFOA/PFOS	LC-MS/MS	Plasma	Median: 39 range: 38–41	ADHD: 132	7
(Vuong, Webster et al., 2021) [44]	America	Median: 28.71	241	PFOA/PFOS/PFHxS/ PFNA	ID-HPLC-MS/MS	Serum	16 ± 3 (*n* = 210) 26 (*n* = 21) 24 h of delivery (*n* = 55)	ADHD: 47	8
(Oh, Bennett et al., 2021) [45]	America	<35 (*n* = 93) ≥35 (*n* = 80)	173	PFOA/PFOS/PFHxS/ PFNA/PC-1	ID-HPLC-MS/MS	Serum	First trimester (*n* = 67), Second trimester (*n* = 142), Third trimester (*n* = 103)	ASD: 57	7
(Lyall, Yau et al., 2018) [51]	America	Mean: 29.9 SD: 5.6 (case) Mean: 28.7 SD: 5.4 (control)	1175	Et-PFOSA-AcOH/Me-PFOSA-AcOH/PFHxS/PFNA/ PFOA/PFOS/PFDeA/ PFOSA	LC-MS	Serum	Range: 15–19	ASD: 553	8
(Skogheim, Weyde et al., 2021) [43]	Norway	Mean: 28.9 SD: 4.98 (ADHD) Mean: 29.5 SD: 4.92 (ASD) Mean: 30.1 SD: 4.45 (control)	2201	PFOA/PFNA/PFDA/ PFHxS/PFHpS/PFOS/ PFUnDA	LC-MS/MS	Plasma	Around 18	ADHD: 821 ASD: 400	9
(Long, Ghisari et al., 2019) [29]	Danmark	Median: 34 range: 17–41 (case) Median: 35 range: 21–44 (control)	210	PFOA/PFOS/PFOSA	LC-MS/MS	Serum	Median: 15 range: 11–20	ASD: 75	9
(Shin, Bennett et al., 2020) [49]	America	Mean: 30.0 SD: 5.8 (case) Mean: 30.7 SD: 5.7 (control)	453	PFOA/PFOS/PFHxS/ PFNA/PFDA/PFUA	HPLC-MS/MS	Serum	Child was 2–5 years old	ASD: 239	9

Abbreviations: ADHD, attention deficit hyperactivity disorder; ASD, autism spectrum disorders; HPLC-MS/MS, high-performance liquid chromatography–tandem mass spectrometry; ID-HPLC-MS/MS, isotope dilution high performance liquid chromatography-tandem mass spectrometry; LC-MS/MS, liquid chromatography-tandem mass spectrometry; NOS, Newcastle–Ottawa Quality Assessment Scale; PFAS, perfluoroalkyl substances; PFOA, perfluorooctanoate; PFOS, perfluorooctane sulfonate; PFHxS, perfluorohexane sulfonate;PFNA, perfluorononanoate; PFHpS, perfluoroheptane; PFDA, perfluorodecanoate; PC-1: selected from the principal component analysis because its eigenvalue was higher than 1; Et-PFOSA-AcOH, N-ethyl perfluorooctane sulfonamide acetate; Me-PFOSA-AcOH, N-methyl perfluorooctane sulfonamide acetate; PFDeA, perfluorodecanoate; PFOSA, perfluorooctane sulfonamide; PFUnDA, perfluoroundecanoate.

**Table 2 ijerph-20-01668-t002:** Summarized odds ratios for ADHD and ASD associated with polyfluoroalkyl substances concentrations in the total concentration group.

Outcomes	Exposures	N Studies	N Participants	OR (95%CI)	P for Z Test	Heterogeneity	
						I^2^	P
ADHD	PFOA	5	2513	0.98 (0.94, 1.02)	0.320	0.0%	0.526
	PFOS	5	2513	0.97 (0.92, 1.03)	0.332	22.1%	0.274
	PFNA	3	995	0.88 (0.75, 1.04)	0.133	49.9%	0.136
	PFHxS	3	995	0.97 (0.91, 1.04)	0.427	0.0%	0.670
ASD	PFOA	5	2035	0.99 (0.89, 1.09)	0.791	42.1%	0.141
	PFOS	5	2035	0.91 (0.77, 1.08)	0.288	56.0%	0.059
	PFNA	4	1825	0.89 (0.70, 1.13)	0.343	51.0%	0.106
	PFHxS	4	1825	1.02 (0.88, 1.20)	0.762	61.3%	0.051

Abbreviations: CI, confidence interval; ADHD, attention deficit hyperactivity disorder; ASD, autism spectrum disorders; OR, odds ratio; PFOA, perfluorooctanoate; PFOS, perfluorooctane sulfonate; PFNA, perfluorononanoate; PFHxS, perfluorohexane sulfonate.

**Table 3 ijerph-20-01668-t003:** Summarized odds ratios for ADHD and ASD associated with polyfluoroalkyl substances concentrations in the highest quartile group.

Outcomes	Exposures	N Studies	N Participants	OR (95%CI)	P for Z Test	Heterogeneity	
						I^2^	P
ADHD	PFOA	5	1267	1.21 (1.04, 1.42)	0.016	42.6%	0.138
	PFOS	5	1267	0.85 (0.721, 0.996)	0.045	0.0%	0.553
ASD	PFOA	4	750	0.85 (0.69, 1.06)	0.159	10.0%	0.343
	PFOS	4	733	0.79 (0.64, 0.98)	0.035	13.3%	0.326
	PFNA	4	733	0.80 (0.65, 0.98)	0.032	0.0%	0.968
	PFHxS	4	784	1.01 (0.71, 1.43)	0.963	62.7%	0.045

Abbreviations: CI, confidence interval; ADHD, attention deficit hyperactivity disorder; ASD, autism spectrum disorders; OR, odds ratio; PFOA, perfluorooctanoate; PFOS, perfluorooctane.

## Data Availability

No new data were created or analyzed in this study. Data sharing is not applicable to this article.

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
