# Peer review of "The Association between Prenatal Per- and Polyfluoroalkyl Substances Exposure and Neurobehavioral Problems in Offspring: A Meta-Analysis"

_ijerph, 2023, doi:10.3390/ijerph20031668_

Round 1
Reviewer 1 Report
Neurodevelopmental disorders have been on the rise for several years. Furthermore, endocrine disruptors represent a booming field of investigation that may be the cause of this increase. The meta-analysis carried out by Yao's team is fascinating work.
Please find below my comments.
The discussion unfortunately remains poor concerning the clinical analysis of the various studies selected.
Methods
Line 98 : “exposure to at least one PFAS”
Line 100: “PFAS exposure during pregnancy using maternal blood samples (e.g. serum, plasma or whole blood)”, but in Ode et al. 2014 PFAS concentration was measured on cord blood!
Line 197: demonstrating a negative correlation does not necessarily imply that PFAS is a “protective factor”
Discussion
Line 342: you say that many studies have different designs and methods but you don’t explain it! What about the relevance of the diagnostic evaluation (questionnaire survey vs expert diagnostis), the patient's testing age (in particular the diagnosis of ADHD at 3 years in Skogheim et al. 2021),...
Line 345 : the discussion of the pathophysiological effect of PFAS is off topic in your article.
Conclusion
In your title, you assess the presence of neurobehavioral problems in the event of exposure to PFAS. However, in the conclusion, you speak of association (whereas it is a question of correlation) which are, moreover, positive and negative. Finally, you stipulate 10 lines before that these results are probably false positives. Be consistent!
Figures
There are many very interesting figures which are not always obvious or correctly annotated. For more readability, I suggest you note on each image the PFAS studied as well as the value of the dose (ng/ml I suppose)?
figures 6, 7, 8, 12 and 13 can be appended so as not to clutter the reading of the results
The figures are not placed in the order of reading. In addition, the table annotations are above while the figure annotations are below, which is disturbing
Author Response
Dear Reviewer,
Thanks for your time giving out such serious comment and precious advice on our manuscript. We have attach great importance to your suggestion which we strictly based on to revise the article. Besides, we have also made some explanation to your question in the attachement.
Again, I am really grateful for your kind guidence. Thank you for your time reviewing our script.

Reviewer 2 Report
1. Significant English editing is required throughout
2. The number of studies that met the meta-analysis criteria is small, and the results are predominantly null apart from 1-2 studies. Therefore, the abstract/conclusions should be revised to reflect greater uncertainty about the relationships studied.
3. The authors place emphasis on the estimated relationship between PFOA and ADHD in the highest quartile and posit that a non-linear relationship exists. However, Figure 10 does not demonstrate a non-linear dose-response relationship in which the effects are large and significant in the highest quartile. Why is this?
4. It is not sufficient to show the results for 1 quartile, the results for all quartiles should be shown
Author Response

(The authors gave the same response as above.)

Round 2
Reviewer 2 Report
Much improved